# Toxic Metal Species and ‘Endogenous’ Metalloproteins at the Blood–Organ Interface: Analytical and Bioinorganic Aspects

**DOI:** 10.3390/molecules26113408

**Published:** 2021-06-04

**Authors:** Tristen G. Bridle, Premkumari Kumarathasan, Jürgen Gailer

**Affiliations:** 1Department of Chemistry, 2500 University Drive NW, University of Calgary, Calgary, AB T2N 1N4, Canada; tristen.bridle@ucalgary.ca; 2Environmental Health Science and Research Bureau, Health Canada, Ottawa, ON K1A 0K9, Canada; premkumari.kumarathasan@canada.ca

**Keywords:** toxic metals/metalloids, chronic exposure, bloodstream, bioinorganic chemistry, metalloprotein, biomarker, mechanism of toxicity

## Abstract

Globally, human exposure to environmental pollutants causes an estimated 9 million deaths per year and it could also be implicated in the etiology of diseases that do not appear to have a genetic origin. Accordingly, there is a need to gain information about the biomolecular mechanisms that causally link exposure to inorganic environmental pollutants with distinct adverse health effects. Although the analysis of blood plasma and red blood cell (RBC) cytosol can provide important biochemical information about these mechanisms, the inherent complexity of these biological matrices can make this a difficult task. In this perspective, we will examine the use of metalloentities that are present in plasma and RBC cytosol as potential exposure biomarkers to assess human exposure to inorganic pollutants. Our primary objective is to explore the principal bioinorganic processes that contribute to increased or decreased metalloprotein concentrations in plasma and/or RBC cytosol. Furthermore, we will also identify metabolites which can form in the bloodstream and contain essential as well as toxic metals for use as exposure biomarkers. While the latter metal species represent useful biomarkers for short-term exposure, endogenous plasma metalloproteins represent indicators to assess the long-term exposure of an individual to inorganic pollutants. Based on these considerations, the quantification of metalloentities in blood plasma and/or RBC cytosol is identified as a feasible research avenue to better understand the adverse health effects that are associated with chronic exposure of various human populations to inorganic pollutants. Exposure to these pollutants will likely increase as a consequence of technological advances, including the fast-growing applications of metal-based engineering nanomaterials.

## 1. Introduction

Ever since Earth came into being 4.5 billion years ago, ‘Panta rhei’ (attributed to Heraclitus), which means ‘everything flows’, has described the dynamic mingling of chemical elements and their species between the geosphere, the hydrosphere, the atmosphere and the biosphere (i.e., all living organisms). Owing to three revolutions—the agricultural, the cognitive and the industrial—Homo sapiens has since multiplied to an extent that all anthropogenic activities cumulatively perturb the global biogeochemical cycles of at least 11 potentially toxic chemical elements [1]. This development has necessitated biogeochemists to add the ‘anthrosphere’ as another environmental compartment to accurately describe the dynamic cycling of chemical elements on the surface of our planet and has—quite appropriately—prompted some scientists to claim that we live in ‘the Anthropocene’, defined as a geological age in which human activity has been the dominant influence on climate and the environment. Certain anthropogenic activities, such as the metallurgy industry, are known to dramatically affect ecosystems [2]. In northern Siberia, for example, 24,000 km^2^ of Taiga (roughly 1/3 of the size of Austria) represent the most polluted area in the world where there is essentially no tree growth due to the emission of 1.8 million t of pollutants (98% SO_2_ as well as Cu and Ni) in 2018 [3]. Anthropogenic inorganic pollutants that are released into the global environment due to technological advances and fossil fuel consumption can also enter the biological system by different exposure routes to adversely affect human health. One of the main routes by which inorganic pollutants are absorbed is the gastrointestinal (GI) tract. Stretched out, the GI tract of a human adult is 8 m long and its gut mucosa covers a surface area of 260–300 m^2^, which corresponds roughly to the area of a tennis court. The GI tract mediates the efficient absorption of all nutrients, including proteins, vitamins, carbohydrates, fatty acids and essential trace elements (e.g., Cu, Fe, Zn) [4], from our diet into the systemic blood circulation for subsequent distribution to organs to maintain health and well-being [5]. Ultimately, however, the GI tract is a double-edged sword as it will also absorb bioavailable pollutants from our diet [6,7] and transfer them into the bloodstream to various degrees [8]. While organic pollutants, such as polycyclic aromatic hydrocarbons (PAHs), certainly need to be considered as well, the primary focus of this paper will be on inorganic pollutants, including environmental metals and organometallic species, such as As^III^, As^V^, Cd^2+^, CH_3_Hg^+^, Hg^2+^ and Pb^2+^ to which essentially all humans are chronically exposed [8], including children, pregnant women and the elderly, who are particularly vulnerable populations [9]. Toxic metals and metalloid compounds, once introduced into the environment, are fundamentally different from organic pollutants in that they cannot be degraded and they can therefore linger for hundreds of years [10]. Internalized toxic metals can access target organs by molecular mimicry, participate in redox cycles (e.g., transition metals), contribute to oxidative stress and form DNA and/or protein adducts, which can have considerable adverse implications for the organism [11,12].

In the biosphere, an important biological compartment where molecules literally move continuously is the bloodstream, which from an inorganic perspective includes nutrient metals (e.g., Cu, Fe, Zn) *and* environmental metal pollutants (As^III^, As^V^, Cd^2+^, CH_3_Hg^+^ Hg^2+^ and Pb^2+^). Crucially, the bloodstream needs to maintain a constant flux of energy and essential trace element ‘building blocks’ for the continuous assembly of biochemically vital metalloproteins within organs (e.g., the liver [13]) and it also receives biomolecular products from tissue/organs [14], which can provide useful information about the health status of an organism [15,16,17]. Despite this, the dynamic exchange of essential *and* toxic metal species at the blood–organ interface is often neglected even though the human bloodstream itself—owing to the presence of up to 10,000 plasma proteins [18], >400 small molecular weight metabolites [19], and oxidative (plasma) and reducing components (red blood cells)—provides a rich ‘playground’ for bioinorganic chemistry processes to unfold at 37 °C. Accordingly, the bloodstream should not only be regarded as a ‘pipe’ which merely mediates the translocation of essential trace elements (beneficial) *and* inorganic pollutants (detrimental) to organs [20], and vice versa since it also represents a biological compartment where bioinorganic chemical reactions of toxicological relevance unfold. The bloodstream must therefore be considered in the development of physiologically based pharmacokinetic models [4,21,22] (Figure 1). Likewise, the intravenous administration of highly cytotoxic anticancer active metallodrugs is of toxicological relevance and the interested reader is referred to a recent review [23]. Given the inherent complexity of the bloodstream, the number of inorganic pollutants of interest and the large variety of biomolecules which may be adversely affected [24], two toxicology-related research goals can be identified at the blood–organ interface which both rely on the application of advanced analytical methods [25] to determine metalloentities to tackle the biological complexity [21]: one which has a clear analytical flavor and another one with a strong bioinorganic one.

The analytical problem refers to the quantification of ‘endogenous’ metalloproteins, which are proteins that contain one or more bound essential metal/metalloid and are present in blood plasma (e.g., transport proteins) as well as RBC cytosol. Just like several other plasma proteins that are routinely quantified for diagnostic purposes [26], the quantification of metalloproteins has inherent diagnostic value [25,27,28] and can be useful in diagnosing genetic as well as non-genetic disorders [18]. At the same time, metalloprotein concentrations in plasma and RBC cytosol may be potentially useful as indirect indicators of the cumulative environmental exposure to individual or multiple inorganic pollutants.

The bioinorganic problem refers to the in vivo formation of metalloentities in plasma and/or RBC cytosol which contain an essential metal *and* a toxic metal/metalloid [29]. While the formation of these species is of inherent toxicological relevance, not all of these species necessarily have a half-life in the bloodstream that qualifies them as potential exposure biomarkers [30]. Nevertheless, knowledge about their transient formation in the bloodstream is crucial as these bioinorganic processes can affect the secretion of metalloproteins from organs to the bloodstream, which therefore directly relates to the analytical problem of quantifying endogenous metalloproteins in plasma. In addition, information on the stability of bioinorganic complexes in different biological compartments is of importance in testing for associations between pollutant exposure and adverse health effects in epidemiological studies in order to choose the proper biological compartment for the appropriate exposure period (e.g., acute or chronic exposure).

Since considerable progress has been made in the quantification of metalloentities in the bloodstream, it appears timely to assess their utility as potentially useful ‘read-outs’ to gain insight into the inherently complex interaction between mammalian organisms and their environment (Figure 1). After a brief overview of the endogenous metalloproteins which are present in human blood plasma and RBC cytosol (i.e., lysate), we will identify the biochemical processes by which chronic exposure to inorganic pollutants may result in increased/decreased metalloprotein concentrations in plasma and/or RBC lysate. We will not elaborate on the available instrumental analytical approaches that can be employed for their determination as this information is available in several comprehensive reviews [25,27,28,31,32,33,34]. Thereafter, we will identify bi- and trimetallic complexes which contain essential *and* toxic metals and are formed in blood plasma and/or RBC cytosol. We will also briefly elaborate on their bioinorganic mechanisms of formation. Last but not least, we will discuss the important role that dynamic interactions between plasma proteins and small molecular weight (SMW) metabolites play in the disposition of toxic metal species to target organs. Systems toxicology will be identified as a useful approach to obtain deeper insight into how the chronic exposure of human populations to toxic metals is potentially linked to disease processes [35].

## 2. Metalloproteins Containing a Single Metal as Biomarker

### 2.1. Blood Plasma/Serum

Although plasma and serum are readily available biological matrices for biomarker discovery, the dynamic range of the plasma protein concentrations makes the detection of low-abundance proteins, which have great diagnostic potential, often quite difficult. Conversely, approximately 10 metalloproteins that contain transition metals are present in plasma/serum (Table 1), which can be readily determined [36].

The Cu-containing ceruloplasmin (Cp, 132 kDa) and the highly abundant Fe-containing transferrin (Tf, 79.7 kDa) have long been known to be integral constituents of these biological fluids for >60 years [38,39]. Both metalloproteins are secreted from the liver and represent key players in the tissue homeostasis of the corresponding transition metals [40,41,42]. It is therefore not surprising that Cp and Tf are routinely quantified in plasma/serum in clinical biochemistry laboratories using a variety of immunoassays, immunoturbidometric or other inexpensive methods. The concentration range of Cp in plasma in healthy adults, for example, is 0.20–0.60 g/L, and while higher values may be attributed to nine possible causes, including rheumatoid arthritis and leukemia, lower values are associated with nine other possible causes [26], including Wilson’s disease [43] and Menke’s disease [44]. This example illustrates that the plasma concentrations of individual metalloproteins are useful in revealing ‘systemic biochemical perturbations’ in an organism, but it has also been pointed out that the quantification of only a single metalloprotein (i.e., Cp or Tf) is often inconclusive in terms of gaining insight into the essential trace element status of an organism [45].

In contrast, the simultaneous quantification of several endogenous plasma/serum metalloproteins which contain Cu, Fe and Zn of an organism would effectively correspond to a ‘snapshot’ of the dynamic equilibrium of each of these transition metals in the environment (i.e., the diet)–bloodstream–organ system [46] and thus effectively provide insight into the homeostatic regulation of each of these transition metals [47]. The ‘snapshot’ approach would therefore provide a more comprehensive measure of the trace element status of an organism as the analysis of blood plasma would allow one to observe increased and/or decreased plasma concentrations of individual metalloproteins pertaining to one transition metal (i.e., Cu, Fe and Zn), provided that reference concentration ranges for each detectable metalloprotein in healthy adults are available. Importantly, the capability to determine multiple metalloproteins of one particular transition metal would allow one to clearly distinguish between a diet-related nutritional deficiency (e.g., all metalloproteins are below their healthy ranges) and toxicologically relevant events, such as a particular toxic metal species-mediated specific inhibition of the assembly of a specific metalloprotein at the blood–organ interface (e.g., the concentration of one metalloprotein is below, but all others are within the healthy range). Although there is experimental evidence that individual plasma metalloproteins have potential as disease biomarkers in humans [48,49,50,51,52,53], their full potential for the diagnosis of neurodegenerative diseases has not been realized [54]. This fact is somewhat surprising given that the concentration of Cu, Fe and Zn in brain tissue is in the mM range [55], which makes the possibility to detect potential neurodegenerative disease-related metalloprotein signatures in blood plasma imminently feasible [56,57].

With regard to relevant metalloproteins in blood plasma ~10 major Cu-, Fe- and Zn-containing metalloentities including Cp, Tf, the Zn-protein α_2_ macroglobulin and the extracellular Cu, Zn superoxide dismutase (ex-Cu/Zn SOD) have been observed (Table 1) [18,36]. Moreover, matrix metalloproteinases (MMPs), which contain Zn in the catalytic site, have been reported in plasma [58,59]. MMPs are a class of multidomain Zn-dependent endopeptidases with different isoforms and are extracellular matrix modifiers. The activity of the various MMP isoforms has been used to distinguish health vs. disease status (e.g., cardiovascular, lung, infection, inflammatory, cancer) and also in pregnancy, and this set of markers has clinical value. Mn SOD, which is found mainly in mitochondria, has also been quantified in human plasma [60]. In addition, four selenoproteins have been detected in mouse plasma [61]. While some researchers have reported that the analysis of excised organs (e.g., the liver) from sentinel organisms that inhabit a particular area is useful in detecting toxic metal-containing exposure biomarkers [62], comparatively less is known about metal-containing biomarkers in the bloodstream, which is in constant contact with all organs. Based on these facts, it seems prudent to critically assess the inherent potential that metalloentities that are contained in plasma and/or RBC cytosol can offer in the context of assessing the chronic exposure of mammalian organisms, including humans, to inorganic environmental pollutants. To this end, hemoglobin adduct formation for the toxic metalloid species AsH_3_ has been previously reported [63]. Metabolomics approaches represent an alternative means to gain insight into the organ damage that is associated with the chronic exposure of mammals to toxic metalloid species [16], and proteomics approaches have similarly been used to identify maternal blood biomarkers to better understand the relationship between the chronic exposure of expecting mothers to toxic metal species and adverse pregnancy outcomes [64,65] (Figure 2A). The determination of endogenous metalloproteins (Figure 2B,C) and metabolites that contain essential and toxic metals in the bloodstream (Figure 2D), however, offers another useful approach to observe inorganic pollution-related perturbations at the systems level and to possibly identify metalloentities that could be used as indirect inorganic pollutant biomarkers because the metalloproteome is inherently less complex than the plasma proteome [66].

#### 2.1.1. Decreased Plasma Concentrations of Individual Metalloproteins

The absence of individual metalloproteins from blood plasma can be indicative of genetic diseases. Wilson’s disease (WD), for example, can be diagnosed by observing negligible Cp concentrations in plasma [40] owing to the decreased assembly of this metalloprotein in the liver and its concomitantly reduced secretion into the systemic blood circulation. In an analogous manner, the chronic exposure of mammals to inorganic pollutants could similarly result in a gradual decrease of the plasma concentration of a particular metalloprotein over time since essentially all plasma metalloproteins are assembled in and secreted from the liver. A decreased plasma concentration of a particular metalloprotein could therefore be caused by a reduced translocation/flux of the corresponding essential element building block into the liver either by it forming a complex with a toxic metal in the bloodstream [67] or by it adversely affecting a molecular transporter that mediates the influx of a particular transition metal ion to the liver [68] (Figure 2B). In addition, the chronic exposure of mammals to one or more inorganic pollutants could selectively inhibit the biosynthesis of a particular metalloprotein within the liver similar to what has been observed in microbes [69,70]. Although cytosolic metallothioneins play an important role in sequestering certain toxic metals (e.g., Hg^2+^) within organs [71,72], some toxic metal species may target biomolecular chaperones that are critical for metalloprotein assembly [73] or otherwise interfere with a metalloprotein’s biosynthesis [74]. All of these aforementioned scenarios would decrease the expression of the corresponding metalloprotein over time with a concomitant decrease of its plasma concentration [18] until it is below the healthy control range [26]. Although this principal approach represents an indirect means to assess exposure, it offers two important advantages. Firstly, the quantification of plasma metalloproteins is inherently more practical than the more invasive approach to detect inorganic pollution-related metalloprotein biomarkers, such as Cd-metallothionein in liver biopsy samples [62] as blood plasma is readily accessible. Secondly, if multiple toxic metal species target the assembly of one particular metalloprotein, the decrease of its metalloprotein concentration in plasma would effectively represent an integral/cumulative measure of the exposure to all toxic metal species. Plasma selenoproteins appear to be interesting candidates to test this hypothesis since Hg^2+^ and As^III^ have been shown to specifically target selenoprotein synthesis [29] and plasma selenoproteins can be routinely quantified [75]. Despite this, only a few studies have reported which demonstrate that the chronic exposure of mammals to a specific toxic metal species is associated with a temporal decrease of the concentration of a specific selenoprotein in plasma [76].

#### 2.1.2. Increased Plasma Concentrations of Individual Metalloproteins

Two Fe-containing metalloproteins are released from tissues into the systemic blood circulation, namely ferritin (Ft), which is constantly released from endothelial and organ cells [77], and myoglobin, the plasma concentration of which is elevated after a heart attack as it is released from heart cells [78]. In an analogous manner, one may envision that the chronic exposure of a mammal to inorganic pollutants could result in organ damage which is associated with the release of metal-based tissue leakage products into the bloodstream. An important prerequisite for the use of Fe metalloproteins as indicators of exposure to inorganic pollutants is that only those metal-containing tissue leakage product(s) are of practical use that have an appropriate half-life and can be detected in the presence of the classical plasma metalloproteins (i.e., Cp, Tf).

The suicidal death of RBCs is referred to as eryptosis, which can be triggered by toxic metal/metalloid species including As^III^, Cd^2+^ and Hg^2+^ [79]. Thus, the rupture of RBCs also represents a potential source of metal-based ‘tissue’ leakage products in the blood plasma. To this end, a major plasma Fe-metalloprotein in human plasma was identified as a haptoglobin (Hp)–hemoglobin (Hb) complex [37]. This Hp–Hb complex is formed in plasma after RBCs rupture and the released cytosolic Hb readily reacts with the plasma protein Hp [37,80]. A disease that directly relates to this observation is a rare human disorder called paroxysmal nocturnal hemoglobinuria (NPH), which is associated with the increased rupture of erythrocytes during the night and afflicts about 0.5–1.5 per million people [81]. Since many inorganic pollutants are able to enter RBCs [82] over their lifetime of approximately 120 days, it is conceivable that chronic human exposure to these pollutants could destabilize RBC membranes and eventually result in their rupture [83]. This, in turn, would result in an increased plasma concentration of Hp–Hb complexes (Figure 2C) provided that their subsequent sequestration from the bloodstream by the spleen and by macrophages is sluggish. Based on these considerations, it is evident that more basic research needs to be conducted before the full potential of endogenous plasma metalloproteins as pollution biomarkers in humans as well as sentinel mammalian organisms can be realized [18]. In support of this notion, an association between maternal lead levels and increased MMPs in plasma has been observed [59].

### 2.2. Red Blood Cells

RBC cytosol contains the Fe-containing hemoglobin (Hb, 64 kDa) and catalase [84], the Zn-containing carbonic anhydrase (CA, 30 kDa), a Cu- and Zn-containing superoxide dismutase (SOD) [85] and an unidentified Mn-containing protein, possibly a Mn-containing SOD [85]. RBCs are assembled in the bone marrow and then released into the bloodstream where they can circulate for about 120 days. Therefore, the determination of Cu, Fe and Zn metalloproteins in RBC cytosol [86] could provide an excellent means to assess the chronic exposure of humans to inorganic pollutants, which accumulate therein [87]. For example, it is known that chronic lead (Pb^2+^) poisoning in mammals targets the heme biosynthetic pathway by inhibiting ferrochelatase, which inserts Fe into heme [88]. Accordingly, chronic exposure to Pb^2+^ eventually results in the formation of a Zn-protoporphyrin within RBCs [89]. The diagnostic relevance of determining the Cu, Fe and Zn metalloproteome ‘snapshot’ is, therefore, immediately apparent as a decreased Hb concentration and an increased cytosolic concentration of Zn-protoporphyrin (it has a different MW than CA) would allow one to diagnose chronic human exposure to Pb^2+^. Since many toxic metal/metalloid species are absorbed by RBCs through a variety of uptake mechanisms [68], the quantification of the Cu-, Fe- and Zn-containing metalloproteins in RBC cytosol has inherent potential to be of use in harboring information about the chronic exposure of a mammal to inorganic pollutants. Another lead-binding protein in the erythrocyte is delta-aminolevulinic acid dehydratase (ALAD), which is known to be inhibited by Pb exposure [90].

## 3. Bi-/Trimetallic Complexes Which Contain Essential and Toxic Metals as Biomarkers

### 3.1. Blood Plasma/Serum

Blood represents the first biochemically complex biological fluid that the inorganic nutrients (i.e., essential trace elements, e.g., Se) and pollutants (e.g., As, Cd and Hg) that are absorbed from the GI tract encounter. Direct experimental evidence for the formation of complexes in plasma which contain an essential and a toxic metal dates back to the mid-1970s when a Hg- and Se-containing complex with a molar ratio of 1:1 was observed [91]. The detection of a related Cd and Se complex (Figure 2D) with a similar stoichiometry suggested that the formation of both complexes is mechanistically related [61,92]. Subsequent studies employed X-ray absorption spectroscopy to structurally characterize the Hg-Se species which was formed in blood plasma 25 min after New Zealand white rabbits were intravenously injected with Hg^2+^ and sodium selenite (Se^IV^) as a (Hg-Se)_100_ species [93]. Interestingly, up to 30 of these Hg-Se nanoparticles bound to the plasma protein selenoprotein P [94]. The formed (Hg-Se)_100_–selenoprotein P adduct is stable in solution at pH 7.4 [95] and is likely the species that binds to the surface of RBCs [96] and is eventually deposited in the kidneys and the liver [97]. Thus, the (Hg-Se)_100_–selenoprotein P adduct fulfills all basic requirements for a useful biomarker [75], but its usefulness as an exposure biomarker for Hg^2+^ exposure requires studies to determine its half-life in the bloodstream. Seemingly unrelated to this it has been reported that forests are sinks for Hg and forest fires therefore release significant quantities of Hg that result in an enhanced Hg accumulation in fish [98]. Since fire fighters are exposed to Hg species when fighting forest fires and considering that they are known to suffer from distinct occupational diseases [99], it would be useful to establish a biomarker for their cumulative Hg exposure which can serve as a basis to estimate their exposure to other, more concerning pollutants. While the detection of (Hg-Se)_100_-selenoprotein P adducts in plasma could therefore be a useful biomarker to assess the exposure of firefighters to Hg^0^ and Hg^2+^, its formation in the bloodstream also implies a decreased influx of the essential trace element selenium into organs, which is likely to gradually decrease the assembly of selenoproteins therein. Thus, human exposure to Hg^2+^ and/or Hg^0^ will eventually result in a decreased secretion of selenoproteins into the systemic blood circulation.

### 3.2. Red Blood Cells

The trace element antagonism between As^III^ and Se^IV^ that was discovered >80 years ago in mammals was eventually demonstrated to be based on the formation of the seleno-bis (*S*-glutathionyl) arsinium ion [(GS)_2_AsSe^−^] within RBCs after arsenite (As^III^) and selenite (Se^IV^) were added to RBC lysate [100]. Interestingly, the addition of CH_3_Hg^+^ to RBC cytosol which had been spiked with (GS)_2_AsSe^−^ resulted in the formation of the trimetallic species (GS)_2_AsSeHg-CH_3_ [101]. The analysis of this RBC cytosol by liquid chromatography, however, revealed (GS)_2_AsSeHg-CH_3_ to be rather unstable which makes its formation not a feasible biomarker. More recently, the addition of Hg^2+^, methylmercury (CH_3_Hg^+^) or thimerosal (it contains CH_3_CH_2_Hg^+^) to RBC cytosol revealed the formation of hemoglobin (Hb) species with distinct Hb–Hg bonds [86]. While the formation of these species in RBC cytosol has direct toxicological relevance as it was recently demonstrated that the binding of CH_3_CH_2_Hg^+^ to Hb will adversely affect the binding of O_2_ [102], their detection represents a convenient means to detect human exposure to the aforementioned Hg species.

## 4. Dynamic Interactions between Toxic Metal Species, Plasma Proteins and SMW Thiols

When toxic metal species enter the human bloodstream, they encounter not only thousands of proteins [14] but also over 400 small molecular weight (SMW) molecules and metabolites, including amino acids, peptides, fatty acids and nucleotides that are present at µM concentrations [19]. The variation in concentration of these biomolecules in the bloodstream has been demonstrated to be, in part, genetically determined [19] and is therefore possibly involved in the distribution of toxic metal species to target organ tissues [103]. Thus, the biological fate of toxic metal species at the blood–organ interface appears to be critically determined by their interactions with plasma proteins and SMW molecules/metabolites (Figure 3).

One SMW metabolite that could be an important player in the toxicology of metals is homocysteine (hCys), which is an intermediate metabolite that is formed by the de-methylation of methionine [105], and is present in blood plasma of healthy adults at concentrations of 8–12 µM [106]. Hyperhomocysteinemia is a disease that is characterized by elevated levels of hCys in blood plasma (e.g., >15 µM) and has been linked to the development of cardiovascular disease, stroke, and Alzheimer’s disease (AD) [105], potentially due to the accidental incorporation of hCys into protein structures instead of methionine during translation that subsequently causes protein damage/aggregation, a hallmark of many diseases. Evidence in support of hCys playing an important role in the toxicology of metals comes from in vivo studies [104] and in vitro studies, which have demonstrated that 500 µM of hCys in the phosphate buffered saline (PBS) mobile phase was able to abstract Cd^2+^ from HSA when a HSA–Cd complex was injected onto a SEC column [107]. Although this hCys concentration was higher than what is observed in hyperhomocysteinemia patients, the fact that the results were obtained under near physiological conditions provide a rationale to more systematically investigate the role that this metabolite may play in the translocation of this and other toxic metal species to target organs. Accordingly, an important research avenue is to connect the bioinorganic chemistry of toxic metal species in the bloodstream with uptake mechanisms on the surface of target organ cells and the biomolecular mechanisms of organ damage [35,108]. This strategy will require not only to determine the structure of the actual metal species that is uptaken into an organ [109], but also to obtain knowledge about their uptake into various organ cell types. Taken together, the integration of the bioinorganic chemistry that unfolds at the blood–organ interface has the potential to establish the mechanisms which may causally link the chronic exposure of humans to toxic metal species with adverse health effects and possibly also to the etiology of diseases.

One toxic metal species that is of considerable public health relevance is the neurotoxin methylmercury (CH_3_Hg^+^), which has a strong affinity for thiol groups (-SH) on proteins and SMW thiols, including homocysteine (hCys) [110]. The competitive interaction of this toxic metal species with thiol groups located on proteins and SMW metabolites, however, is highly dynamic [111] and less well understood [112], but could play an important role in its translocation from plasma proteins across the blood–brain barrier (Figure 3A). To this end, it has been demonstrated that nearly 90% of the CH_3_Hg^+^ located in organ tissues of mice can be mobilized to urine by the oral administration of N-acetyl-L-cysteine (NAC) [113,114]. Studies that are designed to probe the interplay between CH_3_Hg^+^, plasma proteins, and SMW thiols under near-physiological conditions could shed important new light not only on the biomolecular mechanisms that are involved in the translocation of this metal species across the blood–brain and the placental barrier (Figure 3A and Figure 4), but also with regard to how the urinary excretion of CH_3_Hg^+^ may be deliberately enhanced in susceptible populations [115], such as the Inuit which consume comparatively large daily doses of this toxic metal species (Figure 3B) [116]. Probing the interactions between other toxic metal species, plasma proteins and SMW metabolites (e.g., hCys) in the bloodstream is also crucial to gain insight into the biomolecular mechanisms which deliver neurotoxic metal species to the brain [117] (Figure 4), where they may undergo toxicologically important biotransformations [118] and may initiate the apoptosis of astrocytes [119]. Establishing all sequential bioinorganic processes that unfold in the blood–organ system to facilitate the uptake of a toxic metal species into the brain with the biomolecular mechanisms which cause cell damage therein [120] thus offers the prospect of causally linking human environmental exposure to toxic metal species with the etiology of neurodegenerative diseases [121], including Alzheimer’s disease [122].

## 5. Conclusions

It is estimated that the emission of pollutants globally caused 9 million deaths in 2015 [127] and poses a significant burden on the global economy [128]. Even though a variety of efforts are underway to reduce human exposure to toxic metal species [129,130,131], their inadvertent introduction into the food chain is being increasingly recognized [132]. In addition, there is direct experimental evidence that human exposure to toxic metals adversely affects organs [133], pregnancy outcomes [64] and neurodevelopment in children [9]. Given the dynamic flow of inorganic and toxic elements through mammalian organisms and owing to the paucity of information on the mechanistic basis of environmental metal exposure and pregnancy outcomes [59] gaining insight into the environment–bloodstream–organ system [112] is critical. To this end, events that unfold in the bloodstream are of focal interest as they play an important role in determining the onset of organ damage [68,134] and may also be implicated in human diseases of unknown etiology [35,135]. Besides, blood plasma is also a valuable matrix to analyze tissue-specific markers and probably can be useful in screening for both exposure and effect markers in the same compartment simultaneously with the use of appropriate instrumental analysis methods. The analysis of blood plasma and RBC cytosol for endogenous metalloproteins and species which contain essential metal and toxic metals represents a feasible approach to gain insight into the dyshomeostasis of several essential metals *and* to observe toxic metal-containing biomarkers to better assess the health status of an organism. The integration of the bioinorganic chemistry that occurs in the bloodstream with biomolecular processes that unfold in organs emerges as an important research goal to detail our understanding of the environment–blood–organ system to possibly disentangle the environment vs. bad genes dichotomy to causally link human exposure to adverse health effects and possibly diseases [136]. Knowledge regarding the dynamics of the metalloproteome and their potential to serve as plasma/serum biomarkers can be useful in guiding future biomonitoring and epidemiological studies in choosing suitable bioinorganic species in the optimal biological compartment. Understanding the concerted biomolecular events which link exposure to disease is a critical prerequisite before environmental regulations can be further tightened to reduce the emission of toxic metals and metalloids into the environment to ascertain that future generations have the same opportunity to live a healthy life as we do.

## Figures and Tables

**Figure 1 molecules-26-03408-f001:**
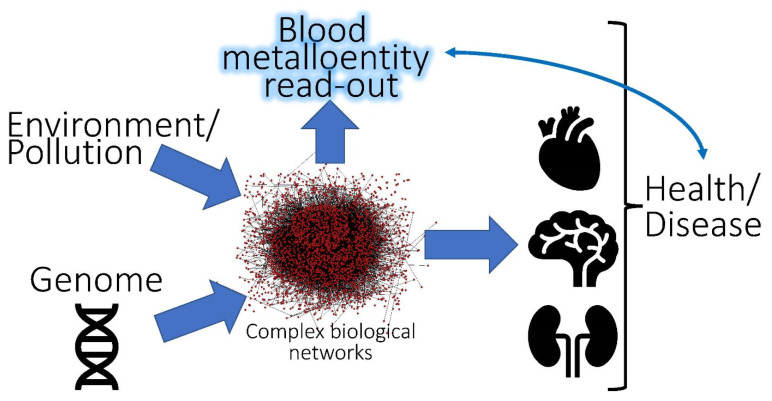
Conceptual depiction of the interaction between the environment and the genome of an organism that involves complex biological networks. The measurements of distinct metalloentities in the bloodstream may serve as biomarkers to assess the chronic exposure of an organism to individual or multiple toxic metal/metalloid species.

**Figure 2 molecules-26-03408-f002:**
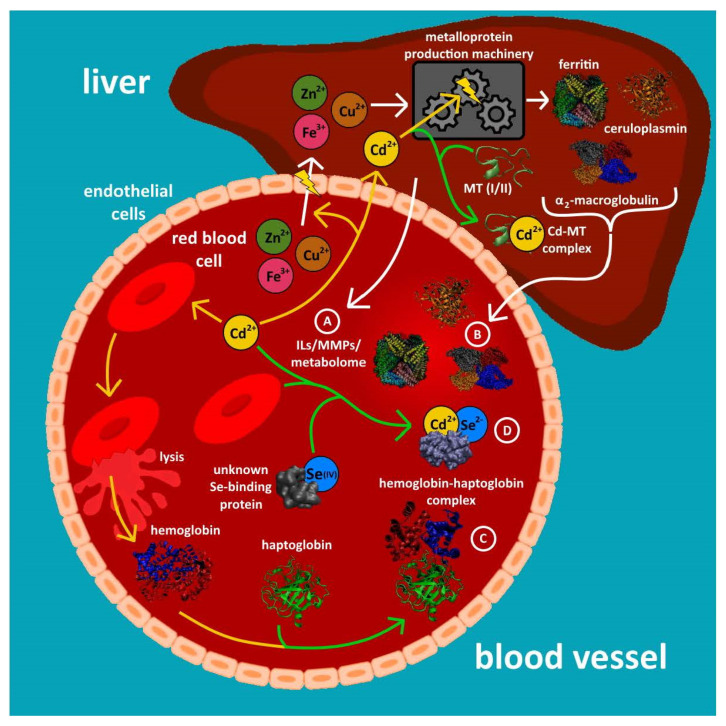
Illustration of the principal mechanisms by which human exposure to toxic metal/metalloid species may result in increased or decreased concentrations of biomarkers in the bloodstream. Mechanism (**A**) refers to increased concentrations of biomarkers that are related to the stress of organs, which includes interleukins (IL) and/or matrix metalloproteinases (MMPs). Mechanisms (**B**–**D**) refer to changes in the concentrations of metalloentities, which includes ‘endogenous’ metalloproteins (**B**), species which contain essential and toxic metals/metalloids (**C**) and a haptoglobin–hemoglobin complex that is formed in plasma following the rupture of RBCs (**D**).

**Figure 3 molecules-26-03408-f003:**
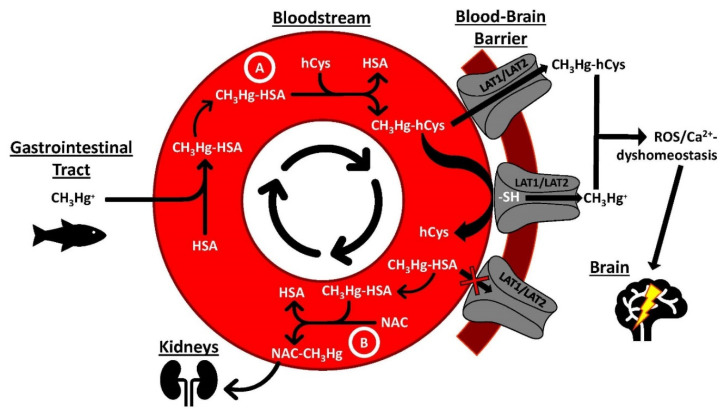
Conceptual illustration of bioinorganic chemistry processes which unfold between CH_3_Hg^+^, plasma proteins and SMW thiols/metabolites in the bloodstream and play a fundamental role in its translocation to the brain and/or its urinary excretion. The mobilization of CH_3_Hg^+^ from human serum albumin (HSA) by hCys may result in the formation of CH_3_Hg-hCys-adducts which may then be translocated across the blood–brain barrier (**A**) by L-type large neutral amino acid transporter (LAT) 1 and LAT 2 [104]. Alternatively, CH_3_Hg-hCys -adducts may donate CH_3_Hg^+^ to a thiol group on transmembrane proteins which ultimately mediate its uptake into the cytosol (**B**).

**Figure 4 molecules-26-03408-f004:**
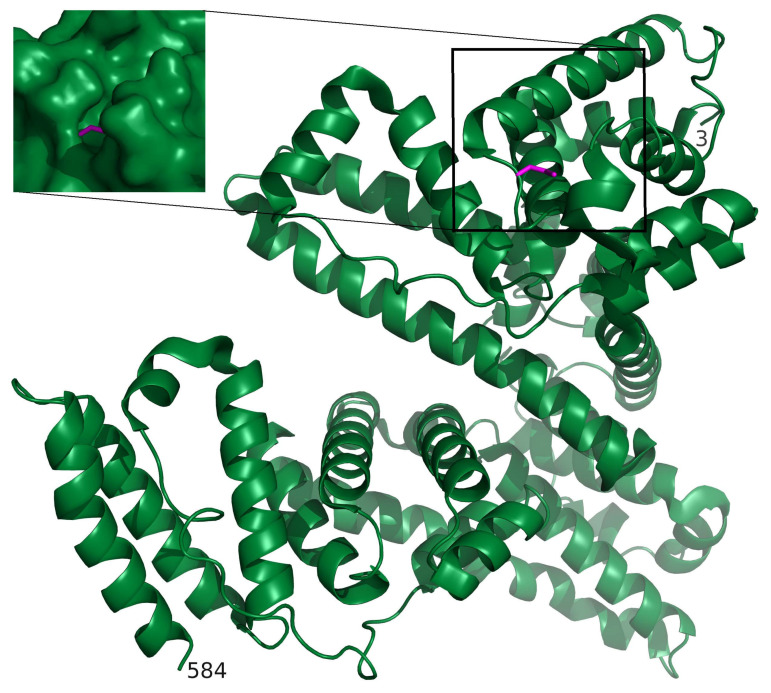
A model of CH_3_Hg^+^ bound to HSA was established by using CH_3_HgCys from the structure of human carbonic anhydrase (PDB identifier 2FOU [123,124]) which was superimposed on Cys-34 of HSA (PDB identifier 3SQJ [125]) using SWISS-MODEL/Swiss-pdbViewer [126]. The ribbon diagram of HSA shows where the binding site of CH_3_Hg^+^ is located and the amino- and carboxy-termini are indicated by residue numbers 3 and 584, respectively. The inset (left) shows a closeup of CH_3_Hg bound to Cys-34 (magenta) with a surface representation of the protein. PyMOL Molecular Graphics System (Version 1.4.1. Schroedinger L.L.C.) was used to establish the structural model. The location of CH_3_Hg^+^ within a cleft suggests that its direct translocation from HSA to LAT1 and/or LAT2 is unlikely and that SMW thiols may critically mediate this transfer.

**Table 1 molecules-26-03408-t001:** Molecular properties of major metalloproteins and metallopeptides in human plasma/serum.

Metal	Metalloprotein or Entity Which Contains Bound Metal	Molecular Mass (kDa)	Number of Metal Atoms Bound per Protein	Reference
Fe	Ferritin	450	<4500	[36]
	Transferrin	79.9	2	[36]
	Haptoglobin–hemoglobin complex	86–900	2	[37]
Cu	Blood coagulation factor V	330	1	[36]
	Transcuprein	270	0.5	[36]
	Ceruloplasmin	132	6	[36]
	Albumin	66	1	[36]
	Extracellular superoxide dismutase	165	4	[36]
	Peptides and amino acids	<5	-	[36]
Zn	α_2_ Macroglobulin	725	5	[36]
	Albumin	66	1	[36]
	Extracellular superoxide dismutase	165	4	[36]

## Data Availability

The data presented in this study are available on request from the corresponding author.

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
