# Peer review of "Toxic Metal Species and ‘Endogenous’ Metalloproteins at the Blood–Organ Interface: Analytical and Bioinorganic Aspects"

_molecules, 2021, doi:10.3390/molecules26113408_

Round 1

Reviewer 1 Report

This review manuscript is weird.  It is well written and describes a lot and a little at the same time.  The introduction is well written.  After this, the next few sections become some related specific details and lots of hypotheses.  The manuscript ends with a vague tying up of these hypotheses.  

No one doubts that metalloentities in the blood can serve as biomarkers.  As the authors touch on numerous ones while hypothesizing on others.  many are long known while others are proposed.  I just do not see what is accomplished overall by the manuscript.  

The manuscript would be strengthened if a table was attached to section 2.1.  Every time a number of items is mentioned, e.g. "12 major Cu, Fe, and Zn-containing metalloentities" these could be listed explicitly in a table.  

I am very uncertain "Molecules" is the correct journal for this manuscript.  I journal with a greater bioinorganic focus would seem much more appropriate.

Author Response

Criticism: This review manuscript is weird.  It is well written and describes a lot and a little at the same time.  The introduction is well written.  After this, the next few sections become some related specific details and lots of hypotheses.  The manuscript ends with a vague tying up of these hypotheses.  

Response: We greatly appreciate this reviewer’s comments who points out that the introduction is ‘well written’, but that ‘the next few sections become some related specific details and lots of hypotheses’. While it is difficult for us to fully understand what this latter statement means, we would like to respond by pointing out that the overall focus of our perspective is to emphasize that a better understanding of the toxicological effects of toxic metal species in humans requires a better integration of the dynamic interchange of metal species and metalloproteins that unfolds at the blood-organ interface. With regard to the criticism that we state ‘lots of hypotheses’ all of our statements are backed with appropriate references from the research literature. To the criticism that ‘the manuscript ends with a vague tying up of these hypotheses’ we respond that our conclusion contains a brief description of the status-quo with regard to the exposure of human to toxic metals followed by a clear recommendation about what kinds of questions future studies should try to address in order to obtain a better understanding of the effects that are associated with the chronic exposure of humans to toxic metal species. 

Criticism: No one doubts that metalloentities in the blood can serve as biomarkers.  As the authors touch on numerous ones while hypothesizing on others.  many are long known while others are proposed.  I just do not see what is accomplished overall by the manuscript.

Response: We agree that the use of metalloentities which contain a toxic metal species can serve as biomarkers, but would like to point out that a large fraction of published studies either evaluate human exposure to toxic metals based total element concentrations of the bloodstream (e.g. studies by the Centers of Disease Control) or by measuring biomarkers in tissues (e.g. cadmium-metallothionein complexes in the kidneys and/or other tissues, which is not at all practical). One of our key arguments is the fact that the temporal decrease of metalloprotein concentrations in human plasma may provide a feasible means to assess chronic human exposure to environmentally toxic metal species provided that their biosynthesis is adversely by the bioinorganic chemistry processes that we outline in our manuscript. We are suggesting that knowledge about the metabolism/biotransformation of toxic metal metabolites in the bloodstream (some of which may have a rather short lifetime therein) is critical to establish which metalloproteins in plasma may serve as complementary and potentially more robust biomarkers to assess human exposure to specific toxic metal species.

Criticism: The manuscript would be strengthened if a table was attached to section 2.1.  Every time a number of items is mentioned, e.g. "12 major Cu, Fe, and Zn-containing metalloentities" these could be listed explicitly in a table.  

Response: We agree with this comment and have incorporated a table (Table 1 on page 8 in revised manuscript) which identifies the main transition metal containing metalloproteins in plasma.

Criticism: I am very uncertain "Molecules" is the correct journal for this manuscript.  I journal with a greater bioinorganic focus would seem much more appropriate.

Response: We were invited to contribute an article to a journal issue with a focus on ‘Molecular and Spectroscopic Insights into Metal Ions Speciation in Extracellular Fluids’. Although we wrote our manuscript with the intent to point to future research avenues/needs, we refer the reader to recent reviews (page 7, references 25, 27, 28, 31-33) that outline advanced spectroscopic tools which need to be employed to make further progress. We would also like to point out that ‘metalloentities’ is a category of biomolecules that fit into the scope of this journal under the theme of ‘chemical biology’.

Reviewer 2 Report

In this manuscript entitled “Toxic metal species and ‘endogenous’ metalloproteins at the 2 blood-organ interface: analytical and bioinorganic aspects” author evaluate principle bioinorganic pro- that contribute to increased or decreased metalloprotein concentrations in plasma and/or 18 RBC cytosol. In my opinion, manuscript require the additional information and explanation before reconsideration for publishing.

Major suggestions the authors might want to consider:

  1. In introduction part should be added discussion about possible molecular implication of metal ions interaction / and or their aqua complexes with protein/metalloproten ( Int. J. Mol. Sci. 2020, 21(6), 2156; and Applied Surface Science 542, 15 March 2021, 148641
  2. What about influence of metal speciation?
  3. Author should extended analytical aspects of work, add some information about e.g. HPLC-ICP MS e.g. in speciation analysis or at least change the title to remove the “analytical”. In this case the sample preparation take the crucial role, e.g. mineralization and implicates artefacts.
  4. Addition of some imaging methods e.g. laser ablation  technique also should briefly mentioned.  

Based on the above review, the manuscript can be recommended for publication after minor revision. If all shortcomings and drawbacks of the current manuscript will be improved, a revised manuscript can be published.

Author Response

In this manuscript entitled “Toxic metal species and ‘endogenous’ metalloproteins at the blood-organ interface: analytical and bioinorganic aspects” author evaluate principle bioinorganic pro- that contribute to increased or decreased metalloprotein concentrations in plasma and/or 18 RBC cytosol. In my opinion, manuscript require the additional information and explanation before reconsideration for publishing.
Major suggestions the authors might want to consider:

Criticism 1: In introduction part should be added discussion about possible molecular implication of metal ions interaction / and or their aqua complexes with protein/metalloprotein (Int. J. Mol. Sci. 2020, 21(6), 2156; and Applied Surface Science 542, 15 March 2021, 148641

Response: We are grateful for the valuable suggestions of this reviewer. In response we have incorporated the suggested papers (references 11 and 12 on page 4 in the revised manuscript) and we also point out that the interaction of metal ions with proteins and or metalloproteins has important biochemical implications (sentence highlighted in red on page 4 of the revised manuscript).

Criticism 2: What about influence of metal speciation?

Response: We have clearly outlined the toxic metal species that our manuscript attempts to address (e.g. AsIII, AsV, Cd2+, CH3Hg+, Hg2+ and Pb2+ on page 4 of the introduction in the revised manuscript) and have always made sure to clearly identify the metal species that is meant in the context of the various mechanisms in our manuscript. We therefore did not see a need to implement any changes in response to this comment. 

Criticism: 3. Author should extended analytical aspects of work, add some information about e.g. HPLC-ICP MS e.g. in speciation analysis or at least change the title to remove the “analytical”. In this case the sample preparation take the crucial role, e.g. mineralization and implicates artefacts.

Response:  Our manuscript was not intended to focus on analytical details (sample preparation and analysis), but rather to point to an important research need (i.e. to better understand the blood-organs interface). To this end we have therefore stated in the introduction (page 7 of revised manuscript) that “We will not elaborate on the available instrumental analytical approaches that can be employed for their determination as this information is available in several comprehensive reviews.” We have, however, included additional references to make the reader aware of important reviews that relevant instrumental analytical techniques (see page 7 in revised manuscript; HPLC-ICP-MS techniques: reference 25, 27, 28, 31; LA-ICP-MS: reference 32, 33 and ESI/MALDI: reference 34).   

Criticism 4: Addition of some imaging methods e.g. laser ablation technique also should briefly mentioned.

Response: We have included an additional review (see reference 33 in the revised manuscript on page 7) that focusses on the application of LA-ICP-MS in the context of measuring metalloentities in biological samples as well as another one that specifically refers to HPLC hyphenated to ESI-MS and MALDI (reference 34). 

Reviewer 3 Report

The manuscript by Brindle et al entitled “Toxic metal species and ‘endogenous’ metalloproteins at the blood-organ interface: analytical and bioinorganic aspects” is aimed to revise the existing literature on the effect of toxic metal species into the blood-serum metalloproteome, as claimed in the title, including the bioinorganic and the analytical aspects. However, I consider that the manuscript fails, in particular, in the second point since there is not analytical information included (in fact, the authors comment in page 4, line 124, “We will not elaborate on the available instrumental analytical approaches that can be employed for their determination as this information is available in several comprehensive reviews”). Therefore, I think that the title should be modified accordingly, in the first place. Furthermore, I do not see any clear structure on the manuscript more than a plethora of research topics that have been covered by the research group authoring the work. The field of metalloprotein analysis in biological fluids (serum or blood) is not such a broad field of research and I consider that the authors have missed important references published on this topic over the years. Therefore, I consider that the manuscript should be seriously modified before consideration for publication in this journal. Some specific aspects are given below:

  1. What is the aim of the work? This is unclear in the introduction since the analytical part is left out. What do the authors try to address? Please, establish the aim of this review.
  2. “2. Metalloproteins containing a single metal as biomarkers” in blood and plasma. How can it be that the authors focus their first paragraphs on ceruloplasmin when there are many more significant fractions binding single metals? For instance, transferrin, with binding sites able to bridge different metal ions and highly abundant (3 mg/mL) is almost neglected here. Albumin, which has the capability to bind many other ions is not even mentioned. I think the work is completely defocused.
  3. “the simultaneous quantification of several endogenous plasma/serum metalloproteins which contain Cu, Fe, and Zn...would provide a snapshot” Where are the references of the work conducted by other authors on the speciation of these elements in serum? And why do they provide a snapshot? At least, in the case of Fe, the absorption through the diet is tightly regulated by the intestine, therefore, I would not agree on the fact that measuring Fe-species in plasma would reveal such information.
  4. MMPs: the authors talk about the importance on the isoforms of the minor protein and they do not mention the possible glicoforms of proteins like hemoglobin or transferrin that truly have a clinical diagnostic value and that can be extremely complex to analyze.
  5. In the same section 2. Metalloproteins containing a single metal as biomarkers (blood and plasma) the authors make reference to “Metabolomics approaches represent an alternative means to gain insight into the organ damage that is associated with the chronic exposure of mammals to toxic metalloid species”. This is totally confusing to the reader. Please, reformulate the whole manuscript.

Author Response

The manuscript by Brindle et al entitled “Toxic metal species and ‘endogenous’ metalloproteins at the blood-organ interface: analytical and bioinorganic aspects” is aimed to revise the existing literature on the effect of toxic metal species into the blood-serum metalloproteome, as claimed in the title, including the bioinorganic and the analytical aspects. However, I consider that the manuscript fails, in particular, in the second point since there is not analytical information included (in fact, the authors comment in page 4, line 124, “We will not elaborate on the available instrumental analytical approaches that can be employed for their determination as this information is available in several comprehensive reviews”). Therefore, I think that the title should be modified accordingly, in the first place. Furthermore, I do not see any clear structure on the manuscript more than a plethora of research topics that have been covered by the research group authoring the work. The field of metalloprotein analysis in biological fluids (serum or blood) is not such a broad field of research and I consider that the authors have missed important references published on this topic over the years. Therefore, I consider that the manuscript should be seriously modified before consideration for publication in this journal. Some specific aspects are given below:

Response: We greatly appreciate the comments of this reviewer. With regard to the criticism that ‘there is no analytical information included’ we would like to point out that we define ‘analytical aspects‘ as the ‘analytical quantification of ‘endogenous’ metalloproteins’, which is different from providing details about the involved analytical procedures. We nevertheless refer the interested reader to important reviews that identify a variety of analytical methodologies (see page 7 of revised manuscript; references 25, 27, 28, 31-34). With regard to the ‘structure of the manuscript’ we first addressed metalloproteins which contain a single metal (in plasma/serum and then in red blood cell cytosol), then metallo- proteins which contain two or three metals and finally how small molecular weight metabolites (e.g. methylmercury) affect the translocation of toxic metal species to organs, including the brain. We believe that this structure clearly summarizes the existing literature landscape for the interested reader.

Criticism: What is the aim of the work? This is unclear in the introduction since the analytical part is left out. What do the authors try to address? Please, establish the aim of this review.

Response: We have stated in the abstract that ‘In this perspective we will examine the use of metalloentities that are present in plasma and RBC cytosol as potential exposure biomarkers to assess human exposure to inorganic pollutants. Our primary objective is to explore principle bioinorganic processes that contribute to increased or decreased metalloprotein concentrations in plasma and/or RBC cytosol. Furthermore, we will also identify metabolites which can form in the bloodstream that contain essential as well as toxic metals for use as exposure biomarkers.’. We believe that this statement sufficiently defines the overall aim of our manuscript and are reluctant to reiterate this statement in the introduction.

Criticism: “2. Metalloproteins containing a single metal as biomarkers” in blood and plasma. How can it be that the authors focus their first paragraphs on ceruloplasmin when there are many more significant fractions binding single metals? For instance, transferrin, with binding sites able to bridge different metal ions and highly abundant (3 mg/mL) is almost neglected here. Albumin, which has the capability to bind many other ions is not even mentioned. I think the work is completely defocused.

Response: We are grateful for this comment. In order to provide important context for the reader we note that while plasma is a readily available biological matrix for biomarker discovery, the plasma proteome has a huge dynamic range and low abundance proteins with relatively great diagnostic potential can be obscured by highly abundant proteins. Albumin, IgG, IgA, transferrin, haptoglobin, and antitrypsin are six highly abundant plasma proteins which are often depleted from the plasma in the study of plasma proteins. We have therefore added an additional sentence highlighted in red on page 8 of the revised manuscript. To further increase the clarity of our manuscript we have incorporated an additional Table which identifies all metalloproteins that are present in blood plasma (page 8, Table 1 in revised manuscript).

Criticism: “the simultaneous quantification of several endogenous plasma/serum metalloproteins which contain Cu, Fe, and Zn...would provide a snapshot” Where are the references of the work conducted by other authors on the speciation of these elements in serum? And why do they provide a snapshot? At least, in the case of Fe, the absorption through the diet is tightly regulated by the intestine, therefore, I would not agree on the fact that measuring Fe-species in plasma would reveal such information.

Response: We have included an additional reference to include recent work by others on the speciation of zinc in serum (see reference 47 on page 9, highlighted in red in the revised manuscript). In response to the question as to what a ‘snapshot’ refers to: an analytical technique that can simultaneously measure several Cu, Fe and Zn metalloproteins in plasma will allow to produce a chromatogram that contains several metal ‘peaks’ which taken together constitute a ‘snapshot’ of the metalloproteome. Since the chronic exposure of humans to toxic metals may affect the intensity of individual metal peaks corresponding certain Cu, Fe, and/or Zn containing metalloproteins (e.g. toxic metals can substitute or influence the synthesis of the proteins), the analysis of plasma/serum for metalloproteins has potential merit to assess chronic exposure to toxic metals. While we agree that the uptake of iron is tightly regulated, we have recently published a manuscript (which we also cite in the manuscript) in which we demonstrate that the analysis of human plasma for iron metalloproteins can provide insight into the rupture of red blood cells and the associated release of hemoglobin (J. Inorg. Biochem. 201, 2019, 110802).

Criticism: MMPs: the authors talk about the importance on the isoforms of the minor protein and they do not mention the possible glicoforms of proteins like hemoglobin or transferrin that truly have a clinical diagnostic value and that can be extremely complex to analyze.

Response: MMPs were incorporated into the manuscript as they are an emerging new class of important biomarkers in various diseases including cancer, cardiovascular disease and adverse pregnancy outcomes and have been studied in the context of environmental exposure-related adverse health effects. We agree with the reviewer’s comment that the glycoforms of proteins such as hemoglobin and transferrin can serve as useful markers, but there is more investigative work that needs to be done for being fit to be applied in environmental toxicology studies. The work described in this manuscript focuses more on tangible markers based on where the field is at right now.

Criticism: In the same section 2. Metalloproteins containing a single metal as biomarkers (blood and plasma) the authors make reference to “Metabolomics approaches represent an alternative means to gain insight into the organ damage that is associated with the chronic exposure of mammals to toxic metalloid species”. This is totally confusing to the reader. Please, reformulate the whole manuscript.

Response: We mention ‘metabolomics approaches’ to acknowledge the work of researchers who use this approach which is complementary to the one that we propose (i.e. metalloentities in plasma as biomarkers).

Reviewer 4 Report

The present paper is a perspective article that propose a proteomic approach for the analysis of metalloproteins in blood as a method to assess the human exposure to toxic metal species. Although it is already well known that changes in metalloproteins levels have an impact on human health, the authors frame these changes in relation to possible environmental pollutants. The paper is interesting.

To aid the reader, before publication, I suggest to the authors to add a table summarizing the main metalloproteins in blood and RBC cytosol, the metal ions they bind, and the known implications for human health, with appropriate references. 

Author Response

The present paper is a perspective article that propose a proteomic approach for the analysis of metalloproteins in blood as a method to assess the human exposure to toxic metal species. Although it is already well known that changes in metalloproteins levels have an impact on human health, the authors frame these changes in relation to possible environmental pollutants. The paper is interesting.

Criticism: To aid the reader, before publication, I suggest to the authors to add a table summarizing the main metalloproteins in blood and RBC cytosol, the metal ions they bind, and the known implications for human health, with appropriate references. 

Response: We have incorporated an additional Table (Table 1 on page 8 of the revised manuscript) which identifies the metalloproteins that contain transition metals and are present in blood plasma.  

Based on the above review, the manuscript can be recommended for publication after minor revision. If all shortcomings and drawbacks of the current manuscript will be improved, a revised manuscript can be published.

Round 2

Reviewer 1 Report

Comments have mostly been addressed.